# Moral Foundational Framing and Its Impact on Attitudes and Behaviours

**DOI:** 10.3390/bs12050118

**Published:** 2022-04-20

**Authors:** Tamim Mobayed, Jet G. Sanders

**Affiliations:** 1Department of Psychological and Behavioural Science, The London School of Economics and Political Science, London WC2A 3PH, UK; j.g.sanders@lse.ac.uk; 2Wolfson College, University of Oxford, Oxford OX2 6UD, UK

**Keywords:** moral foundations theory, refugees, framing, moral framing, behaviour change

## Abstract

This preregistered, randomized controlled experiment attempts to distil the effects of Moral Foundational Framing on attitudes and behaviours towards refugees in the UK. First, moral foundations were found to robustly predict both attitudes and behaviours practised towards refugees. Next, a degree of support was found for the effectiveness of moral foundational framing in adjusting attitudes, but not behaviour, toward refugees in the UK. Individuals who scored highly on certain morals were susceptible to influence by moral foundational framing, but not always in ways that may have been expected. We conclude that the robust relationship found between moral foundations and attitudes towards refugees stresses the importance of actively shaping moral foundations. This notion is strengthened by the comparatively less robust effects of framing. Findings could be used to inform the practice of individuals interested in influencing opinion and behaviour, particularly in support of refugee acceptance.

## 1. Introduction

The world is facing unprecedented challenges in the form of the plight of refugees. 70.8 million people have been forced to leave their homes and live away from them today, of which 25.9 million are refugees [1]. Of this 70.8 million, it is estimated that more than a third were forced to move due to sudden onset weather events [2]. According to the UNHCR, in 2018, there were a total of 126,720 refugees in the UK, including 18,519 who had been offered asylum that year [1]. The Migration Observatory at the University of Oxford reported that more than 50% of British people believe migration should be reduced. Blinder [3] reported that more than 75% of British citizens want to reduce immigration.

Negative attitudes centre around a triad of attitudinal bunches; economic burden, sectarian/ethnic considerations, and safety of the host population. Such attitudes have been reported widely, including within the context of Turkey [4], Japan [5], and Europe-wide [6,7,8]. Buffers against holding negative attitudes towards refugees seem to come from higher levels of education [5,9] while the informational flow from elites also holds sway [5].

Researchers have employed a range of approaches in their attempts to change attitudes towards refugees, including making salient certain social norms [10], holding individuals accountable for decisions they make about other groups [11], employing an influencing confederate [12], or close friends and acquaintances [13], information dissemination that challenges ill-attitudes with facts [5,7,9], addressing specific concerns such as ethnic composition and value worries [14], and consciousness raising [15]. 

### 1.1. Framing in Political Contexts

Framing grew from Prospect Theory; this theory centres on the finding that “decisions taken by individuals can be altered by presenting information in logically equivalent but semantically different ways” ([16], p. 5). Their research found a significant relationship between the way in which information was presented and how it would be weighed by individuals. Framing can be described as a pointed packaging of information that might draw emphasis to one (or certain) aspect(s) of it; crucially, it does not result in any factual changes to the information. 

In politicized decision contexts, Druckman and McDermott identify two key deviations by way of emotions and susceptibility to the framing effect [17]; different emotions, both deemed negative, e.g., anger and distress, can lead to differential impacts of framing. The same authors identified emotions as one potential way to moderate the framing effect. Ref. [18] used conflicting elite discourse to see if it could reduce the impact of the framing effect, with mixed success. Researchers at Stanford University [19] deployed framing to tackle a phenomenon researchers call the ‘progressive paradox’, wherein individuals favour a more egalitarian society yet readily vote against candidates who seek to realize that goal at the ballot box. This built on previous research, such as Feygina, Goldsmith and Jost [20] who reported that they could increase support for environmental policies if they were framed in terms of preserving American traditions.

### 1.2. Moral Foundations Theory

One common approach to framing in these politicized contexts [19,20] is by means of Moral Foundations Theory (MFT) [21]. MFT attempts to formulate a comprehensive theory of human morality, identifying moral values that extend across cultures. By examining a range of cultures, with theoretical foundations in social psychology and anthropology, they extract five (or six [22]) universal moral dimensions: care/harm, fairness/cheating, loyalty/betrayal, authority/subversion and sanctity/degradation, which can be found within all cultures, to varying intensities and differential manifestations (the proposed sixth moral is liberty/oppression). In this study, we look at the effect of four of these frames: care, fairness, loyalty or sanctity. Authority was deselected due to both resource constraints and expectations it would not be influential (though this exclusion proved to be erroneous: statistical analyses on data collected through this study found strong, significant relationships between the moral of authority and some of the outcome measures). Each of the five foundations is rooted in evolutionary benefit, which we outline below.

#### 1.2.1. Care/Harm

This evolutionarily important value is prominent across different mammals, one form of which is embodied by a mother’s care for her offspring; the development of expressions of care is often socially driven by the propagation of normative ideals. While the biopsychological hardware for this foundation might be similar between humans and societies, its manifestation varies greatly. Compare the expression of care/harm in Ancient Spartan society with that of contemporary California. 

#### 1.2.2. Fairness/Cheating

Fairness/Cheating is linked to the notion of reciprocal altruism, where one individual (temporarily) sacrifices for another unrelated individual. Fairness is derived from overarching reciprocal benefit, following the initial cost. Sensitivity to fairness is universal [23] and observed early in human development (before the age of five, and possibly before the age of one).

#### 1.2.3. Loyalty/Betrayal

Loyalty is defined as strong feelings of allegiance toward a person or group and is equally foundational. Both chimps and humans have been found to group together and initiate conflicts with other groups for territory, with humans having intensified this ability due to the development of ‘language, weapons, and tribal markers’ ([21] p. 70). These predispositions towards loyalty were famously activated in Sherif’s [24] seminal study that took a group of schoolboys on a camping trip, divided them into groups and had them face off against each other in competitions; they exhibited commitment to in-group members and stark hostility towards ‘the other’. Indeed, in-group loyalty is often associated with hostile attitudes towards out-group members [25]. 

#### 1.2.4. Authority/Subversion

Dominance hierarchies are seen in many animals, especially larger mammals. Within human beings, social hierarchies vary greatly from culture to culture and might include patriarchy or matriarchy as defining features. Within modern nation states, various cultures might exist (and coexist); liberals might deem obedience to authority as a vice while conservatives champion it as a virtue [26]. 

#### 1.2.5. Sanctity/Degradation

Sanctity and degradation are related to the immune system, physiological aversion and disgust. The evolutionary advantages of this system are great: parasites, bacteria and viruses pose significant threats to our collective survival. Revulsion keeps these (where possible) out of our food, bodies and societies [27]. This system also impacted our social world; “Disgust and the behavioral immune system have come to undergird a variety of moral reactions, for example, to immigrants and sexual deviants” ([21], p. 71).

### 1.3. Moral Framing to Shape Pro-Refugee Attitudes and Behaviour

In this randomized controlled experiment, we deployed moral foundational framing to influence attitudes and behaviours in relation to refugees in the UK. Moral foundational framing has successfully been deployed in contexts including attitudes toward the environmental [20] and fiscal policy [19]. In proximity to this study, Nath et al. [28] applied Moral Foundational framing to immigrant contexts, focusing on the differences between individualizing and binding (The individualizing morals are care/harm and fairness/cheating; the binding morals are loyalty/betrayal, authority/subversion and sanctity/degradation) morals and the way in which these moral groupings predict attitudes. Their experiments successfully applied this form of framing to influence stated attitudes regarding immigration to the USA. Kaufman [14] successfully applied moral framing, showing that reframing discussions around migration could lessen it as a concern. However, most research did not consider how these attitudinal shifts respond to behaviour [16,29]. Grigorieff, Roth and Ubfal [9] found that while certain morally aligned frames changed attitudes, critically, they did not observe behaviourally-inclined changes, such as in decisions made about policy or resource distribution. In the current experiment, we therefore supplement the examination of attitudes by measures approximating pro-refugee behaviours in the real world by asking participants to sign a petition either in favour or against increasing the number of refugees in the UK and inviting them to donate 20% of their experimental earnings to a refugee charity.

Another missing element from the body of literature is the notion of personalization. Studies tend to focus on identifying which one frame may function more effectively than another, or which group responds most to a certain frame. However, increasingly, this one-size-fits-all perspective is being replaced by a more tailored approach [30]. Attempting to understand how moral foundational underpinning might influence individual-level responses is key to the impact of any informational campaign. 

In sum, we expected that Moral Foundations would significantly predict attitudes and behaviours towards refugees. To be more specific, based on patterns observed in the literature, we expected that moral values of care and fairness significantly predict favourable attitudes and behaviours towards refugees. Second, we expected that frames that were congruent and morally aligned would be most effective in eliciting pro-refugee attitudes. Third, we expected that congruent and morally aligned frames will be most effective in eliciting pro-refugee behaviours.

## 2. Materials and Methods

### 2.1. Experimental Design

This study utilised a one-way between subjects’ experimental design with levels (control, care, fair, sanctity and loyalty message frame). In addition, participants provided demographic information (age, gender, racial group and religious affiliation) and political demographic information (political party alignment, political ideology, and position on Brexit). 

### 2.2. Control Group

With the benefit of hindsight, the control group employed by this experiment, when compared to the treatment groups, might have been significantly flawed. Relying on the work of Kaufman [14], the passage used for the treatment groups is arguably contentious and scientifically noisy, evoking themes of inclusion and exclusion relating to migrants and refugees (Appendix B—Table A2). The control group read a significantly different passage that contained none of these themes, leading to a potentially critical difference between the treatment groups and the control groups that was not intended. Accordingly, the control group was excluded from most analyses reported.

### 2.3. Stimuli

We used the Moral Foundations Theory Questionnaire-30 (MFQ-30), a 32-item measure of individual values (30 items, plus two attention checks) to measure participants on their level of five morals. The treatment conditions were devised by way of infusing three written passages with morally laden wording. The passages were threefold: a statement on immigration in the UK (Figure 1), an excerpt from a speech of a fictional politician (Figure 2) and a policy proposal (Figure 3). The treatment groups were five: a control group, as well as four different morally framed conditions. All moral frames were infused with wording that stemmed from each moral value (Table A1 and Table A2). Rather than the statement on immigration, the control group received a passage on the role of lobbyists. While the excerpt from the politician centred on refugees for four of the treatment groups, for the control group, the fictional politician spoke on voting systems. 

### 2.4. Measures

Attitudinal measurement was threefold. Participants were presented with a statement regarding finding the number of refugees entering the UK as being of concern or alarming and were asked to agree (1) or disagree (0) with it. Based on the work of Voelkel and Willer [19] participants were asked to provide the likelihood of their support for a pro-refugee political candidate (on a 5-point Likert scale). Finally, participants were asked to either provide support (1) or not (0) for a policy that would see refugee numbers increase in the UK by 5%. 

The behaviour measures employed were twofold. One involved an invitation to sign a petition in support of, or against, increasing refugee numbers. The second behavioural measure was an invitation to donate 20% of participation in experiment earnings (£0.20) towards a fictional UK-based refugee charity. While this amount might seem insignificant, it represented 20% of the total incentive offered to participants to partake in this study. It can fairly be deduced that for an individual motivated by £1, one-fifth of that amount would carry value.

### 2.5. Participants

1076 British participants (Age: M = 36.8, range: 18–84; 641 females—Table 1 below). All were recruited through the Prolific platform.

### 2.6. Consent

Participants actively consented to participation in the study by way of opting into the experiment and indicating their consent. 

### 2.7. Sample Size

To determine the sample size required to adequately carry out this experiment, a G-power calculation was conducted, using the following parameters:

Effect size (d)—0.35; Power—0.95; Alpha (p)–0.05.

An effect size of 0.35 was selected due to being as close to a strong effect (0.25) as possible, while not exceeding available resources. This yielded *n* of 214 per treatment group, with a total *n* of 1070. 

### 2.8. Recruitment

Participants were recruited in October 2019 using Prolific, deemed a sound recruitment tool that avoids some of the problematic aspects that arise on MTurk [31]. Participants were reimbursed for their time at a rate of £6.43/hour. Participants were instructed they would be ‘*answering a series of questions about their values and attitudes, and taking a number of measures relating to the population of the UK*’.

### 2.9. Ethical Considerations

Participants were informed that they would be free to withdraw from the study at any point. They were also provided with the email address of the LSE’s Research Governance Manager and invited to contact them should any concerns or complaints arise. Participant data was anonymized and all data was treated securely, being accessible only to the researcher and their supervisor. This study was ethically self-approved according to the LSE’s Psychological and Behavioural Science ethical guidelines. 

### 2.10. Procedure

Participants were instructed to complete some of the demographic measures before they were instructed to read the first framed item carefully and state their agreement/disagreement with the first attitudinal measure. They were then instructed to read the second framed item and state their likelihood of voting for the fictional political candidate. Participants then read the framed policy proposal and stated their support or opposition to it. After the attitudinal measures, participants completed the two behavioural measures. Firstly, participants were invited to sign a petition in support of, or a petition against, increasing the number of refugees allowed into the UK. Secondly, participants were invited to donate £0.20, or any other amount, from their participation earnings.

Participants were then asked to carry out the MFQ-30, followed by providing the remaining demographic information. This was purposely set to be done after the main attitudinal and behavioural measures, as it was determined these were the most important measures of the experiment, hence this was done to protect them from participant fatigue. On completion, participants were shown the participant debrief page. 

### 2.11. Exclusion Criteria and Process

The MFQ-30 has two attention-check questions. Q6 comes within the first section, in which participants are determining how relevant the items are to their decision of whether or not something is right or wrong, The item of Q6 states: ‘Whether or not someone was good at math’. Q22, in the second section, in which participants are stating their agreement or disagreement with the items, states: ‘It is better to do good than to do bad’. 

According to the authors of the MFQ-30, any positive response to question six and any negative response to 22 should disqualify the participant. Eight participants violated both conditions and were removed from the data set. 18 participants violated Q22 and were removed from the data set. 159 participants violated Q6. On further enquiry, it was thought that this position could be challenged (This was challenged by two participants who were flagged for potential disqualification). As such, a case could be made that utilitarianism, and consequentialism as a whole, are based on the idea that maths and ethics are related. Accordingly, an individual’s level of maths might be a minor factor in processing morality. A case could thus be made, very much equivocally, that intelligence and ethics can be related; certainly, consequentialist ethics relies on mathematical valuation. In light of this thinking, participants who violated Q6 were only removed if they completed the questionnaire in less than four minutes. 29 participants were removed based on the amended criteria. 

### 2.12. Materials

Stata (15.0); Qualtrics (www.qualtrics.com); Prolific (www.prolific.co).

### 2.13. Pre-Registration

This experiment was pre-registered with the Centre for Open Science (OSF). Areas of focus and expected findings were listed and described here (https://osf.io/7tvj6/). 

## 3. Results

### 3.1. The Relationship between Moral Foundations and Attitudes towards Refugees

In order to probe potential relationships between moral foundations and the combined attitude measures towards refugees, an Ordered Logit regression was executed. The regression analysis, with a Pseudo R2 of 0.089, showed that pro-refugee attitudes were significantly predicted by all moral value measures, barring sanctity (z = −0.26, *p* < 0.80, OR = 0.979 95% CI [0.836–1.147]). Care (z = 4.21, *p* < 0.001, OR = 1.55, 95% CI [1.26–1.90]), fairness (z = 7.32, *p* < 0.001, OR = 2.28 95%CI [1.83–2.85]), loyalty (z = −2.26, *p* = 0.024, OR = 0.81 95% CI [0.666–0.971]) and authority (z = −6.20, *p* < 0.001, OR = 0.527, CI [0.43–0.645]), were all found to be significantly related to attitude towards refugees—fairness and care positively so, loyalty and authority negatively so (See Figure 4A). As predicted, care and fairness are the strongest predictors of positive attitudes towards refugees.

### 3.2. The Effect of Moral Foundational Framing on Shifting Attitudes

To assess the effectiveness of framing conditions on observed attitudes, we ran the same Ordered Logit regression, including the effect of framing conditions in two blocks: first as a main effect and second with interaction effects. The model fit did not significantly improve (X2(4) = 7.36, *p* = 0.118) by adding a main effect (Pseudo R2 of 0.091). The model did significantly improve by adding main and interaction effects (X2(20) = 55.96, *p* < 0.001; Pseudo R2 of 0.101). This model displays a main effect of the care frame (Z = −4.07, *p* < 0.001, OR = 0.007, 95% CI = 0–0.08) and sanctity relative to the control frame (Z = −2.05, *p* = 0.041, OR = 0.14, 95% CI = 0.02–0.09) relative to the control frame.

Despite the unexpected directionality of the main effects (due to previously explained issues with the control group), we also note a number of interaction effects: those who scored higher on authority were less likely to respond to the care (Z = −2.70, *p* = 0.007, OR = 0.4, 95% CI = 0.2–0.78), fair (Z = −2.97, *p* = 0.003, OR = 0.48, 95% CI = 0.2–0.72) or loyalty (Z = −2.89, *p* = 0.004, OR = 0.4, 95% CI = 0.20–0.73) frame than the control. Those who scored higher on loyalty were more likely to respond to the care frame than to the control (Z = 1.986, *p* = 0.047, OR = 1.85, 95% CI =1.01–3.42), which was most consistent with our prediction. Those who scored higher on for care also responded better to the care frame (Z = 2.29, *p* = 0.022, OR = 2.12, 95% CI = 1.12–4.05).

We repeated this analysis without the control group (care frame as a reference). This yielded similar results: there were no main effects of frames if these were added alone but there were when interaction effects were included (Appendix A for the complete analyses). We reject the null hypothesis: not only is no one frame well suited to all participants, but some predispositions are also predictive of the frame type that may direct their attitude toward more prosocial behaviour.

### 3.3. The Relationship between Moral Foundations on Pro-Refugee Behaviour

Did differences carry through to behaviours? We reviewed the relationships between moral foundations and behaviours practised towards refugees—two regressions models were executed. In relation to signing the pro-refugee petition, we ran a Ordered Logit regression (Pseudo R2 of 0.032): fairness was a significant positive [Z = 4.03, *p* < 0.001, OR = 1.62, 95% CI = 1.29–2.06], and authority was a significant negative predictor [Z = −3.56, *p* < 0.001, OR = 0.68, 95% CI = 0.55–838] of pro-refugee signing (see Figure 4B).

In relation to pro-refugee donations, we ran a binomial logistic regression model (Pseudo R2 of 0.096). We found that care [Z = 5.40, *p* < 0.001, OR = 2.04, 95% CI = 1.58–2.65] and fairness [Z = 2.07, *p* = 0.038, OR = 1.33, 95% CI = 1.01–1.73] were significantly positive and authority [Z = −3.75, *p* < 0.001, OR = 0.63, 95% CI = 0.50–0.80] was a significant negative predictor of donating (see Figure 4B).

In sum, we reject the null hypothesis: the foundation of care predicts pro-refugee behaviour, whereas authority does so negatively. Interestingly, differences were less pronounced with monetary donations (odds ratio around 4) relative to the signing of the petition (odds ratio around 2).

### 3.4. The Effect of Moral Foundational Framing on Pro-Refugee Behaviour

To review the effects of framing on pro-refugee behaviour, we added framing conditions to the Ordered Logit model for pro-refugee petitions and the logistic regression model for pro-refugee donating.

For pro-refugee petitions, the predictive power of the model did not significantly improve with added effects of condition (X2(4) = 1.97, *p* = 0.741; Pseudo R2 of 0.033), nor when interaction effects between framing and moral foundation were added (X2(20) = 22.1, *p* = 0.334; Pseudo R2 of 0.044). Upon review of the two models, there were no main interaction effects. Findings did not differ when the control group was removed from the analysis (Appendix A).

For pro-refugee donations, we used the same approach. We also observed no changes in predictive power of the model when the framing condition was added (X2(4) = 4.12, *p* = 0.39; Pseudo R2 of 0.099), nor when interaction effects between framing and moral foundation were added (X2(20) = 22.16, *p* = 0.332; Pseudo R2 of 0.1147). Upon review of the two models, there were no main of interaction effects. For this measure too, findings did not differ when the control group was removed from the analysis (Appendix A).

## 4. Discussion

In this study, we find first that moral foundations significantly predict both attitudes and behaviours practised towards refugees—fairness and care positively so, and sanctity and loyalty negatively so. Second, we find that moral foundational framing was successfully deployed to influence attitudes towards refugees, amongst some groups, most consistently in relation to the care frame and at times for the fairness frame. Third, we find that moral foundational framing had no impact on shifting pro-refugee behaviour.

Our results also demonstrate what type of attitudes are likely to shift by means of moral framing. Presenting individuals with care and fare frames evoked stronger support for voting intention for a pro-refugee political candidate. No such shifts were found for being not concerned over refugee numbers or refugee policy. These findings show that tailoring messaging based on an individual’s morals may hold potential in particular politicized domains [19,20,28]. This impact is weakened by the inability to find significance between framing and two of the three attitudinal measures.

When we take a closer look at the differential effect frames, the morals of care and fairness were associated with positive attitudes towards refugees, while authority was negatively associated. This finding had been reported previously [29]. The replication of this result highlights the level of significance that might be offered by framing. It might serve as a useful influencer of opinion and behaviour; however, the impact of moral values is more robust. Notably, this does not automatically make it useful for shifting attitudes: moral values are not likely to be changed by way of one experiment, but rather, are formed over the course of an individual’s psychological and moral development.

These findings do offer further support for the possible uses of moral foundational framing in garnering support and aiding in persuasion. Critically, support was not found in overturning moral impulses by way of framing; individuals who scored highly in binding values (sanctity and loyalty), who would typically not be forthcoming in support for refugees, were not persuaded by the sanctity and loyalty framed messaging. Our results thus failed to replicate findings by Feygina, Goldsmith and Jost [20], and Voelkel and Willer [19], who were able to shift moral positions based on this form of framing.

The predictive power of the moral foundations (rather than the frame) might offer a candid indication of where effective change lies: attempts at shaping core morals might hold more potential than adjusting the framing they encounter, especially as frames had no impact on behaviour, even if they shifted attitudes. An interesting finding relating to the discrepancy between attitudes and behaviours is found in the study’s secondary focus. When asked if they would support a hypothetical policy that would see more refugees brought into the UK, 65.24% of participants said they would. When they were asked to sign a petition in support of this goal, this number halved to 32.59%, thus again highlighting the discrepancy between changing attitudes and changing behaviours.

The inability of this study to evoke a change in moral applications might of course be due to ineffective deployment of moral framing: the moral frames used might not have been sufficiently congruent. Another possibility is that the loyalty and sanctity-framed texts might have served to prime individuals to feel morals of which they were already in possession. If this were true, it would highlight one of the potential difficulties of using framing in this way. Another possible limitation is the extent to which real behaviour can be captured online. In addition, the sampled population group, as is typical of online populations, was distinctly left-wing, as highlighted by the more than 2:1 ratio of Labour to Conservative voters amongst participants. This might be addressed by means of field experiments. An addressable weakness was identified in the measure that asked participants whether they would support a policy to increase refugee numbers by 5%: could responses to this have been influenced by the differing degrees of what 5% more refugees in the UK would actually mean in real-world numbers? A simple follow-up question could be employed in future to account for this.

The most pressing future avenue of enquiry will continue to be testing how and when framing can be catalysing enough so as to overturn a pre-held belief. This study only showed that framing can increase the strength of attitudes. While this is an important finding in and of itself, were the circumstances under which framing might overturn an attitude discovered, a significantly different range of possibilities will open to those invested in changing attitudes and behaviours towards the oppressed, marginalized and disenfranchised. The robust relationship found between moral values and attitudes towards refugees makes salient the importance of actively shaping moral foundations, in turn shaping attitudes and behaviours towards refugees.

## Figures and Tables

**Figure 1 behavsci-12-00118-f001:**
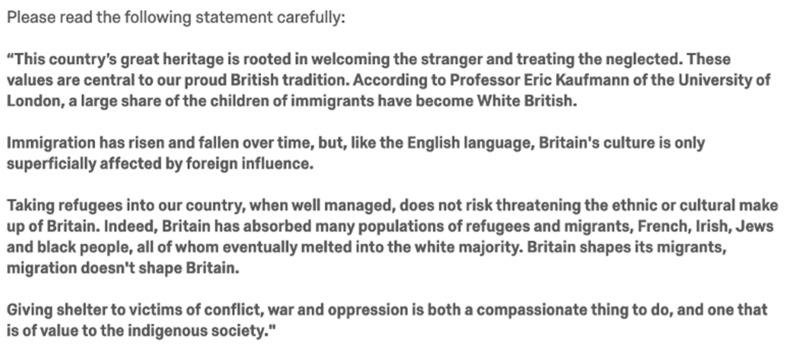
Morally framed passage (care).

**Figure 2 behavsci-12-00118-f002:**
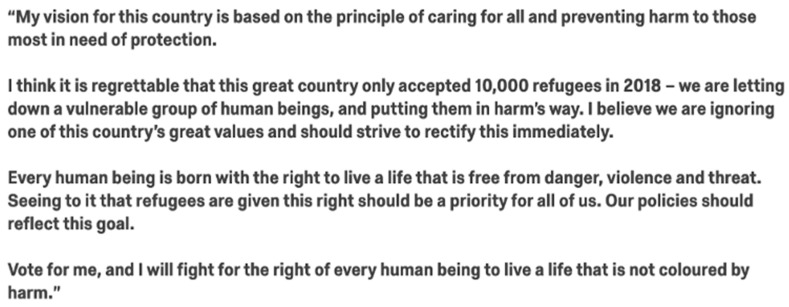
Morally framed political speech excerpt (care).

**Figure 3 behavsci-12-00118-f003:**
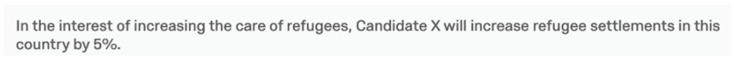
Morally framed policy proposal (care).

**Figure 4 behavsci-12-00118-f004:**
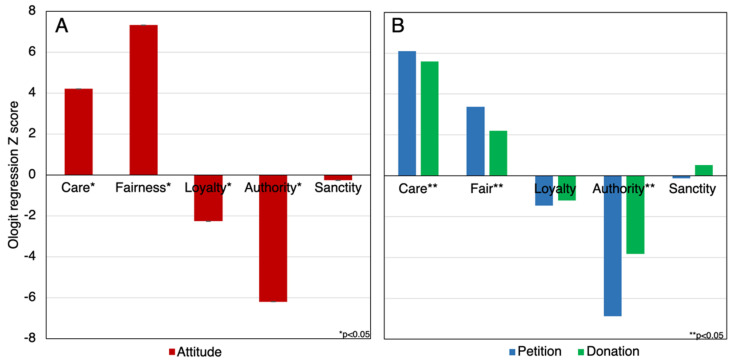
Displaying moral foundation scores by (**A**) the average attitude toward refugees and (**B**) pro-refugee behaviour.

**Table 1 behavsci-12-00118-t001:** Internal Validity of Randomization.

Demographic Information	Care (*n* = 210)	Fair (*n* = 215)	Loyalty (*n* = 217)	Sanctity (*n* = 216)	Control (*n* = 218)
**Male**	83	85	89	82	84
**Female**	126	128	125	132	130
**White**	178	184	189	176	189
**Ethnic Minority**	32	31	28	40	29
**Bachelor’s Degree and Higher**	105	102	104	109	112
**Less Than Bachelor’s**	105	113	113	107	106

## Data Availability

Data collected as part of this study are available on request from the corresponding author.

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
