# Peer review of "Moral Foundational Framing and Its Impact on Attitudes and Behaviours"

_behavsci, 2022, doi:10.3390/bs12050118_

Round 1

Reviewer 1 Report

This paper does not require a lengthy review. It is a well-designed and executed experimental study with a laudable normative objective – namely to understand and improve attitudes towards refugees in the UK.

I believe that studies like this that locate however minor gaps in the literature and our understanding offer much for the current and future needs of refugees in terms of humanitarian acceptance rather than vilifying rejection. More studies need to add to this as part of the scientific enterprise of incremental progression of new knowledge relating to the psychology of attitudes and behaviour towards refugees. The use of Moral Foundation Framing that offers added dimensions than simply racism labelling is a more sophisticated approach that should be explored even further.

This is certainly a publishable paper, although some attention is required to expression, grammar, and punctuation. This can easily be achieved by a thorough copy edit.  For this I can draw your attention to the following lines: 93, 97 (with, not within), 108, 128, 126-7 (needs clarification), 145, 212, 254, 310, 370, 401.

In addition to the above,

  • line 83 makes some distinction “between humans and societies”. Not quite sure what the distinction refers to here.
  • Line 283 and below make references to a hypothesis and null hypothesis. These have not been mentioned before, and nor is it explained what they are (unless I have missed it). They should be made clear in the discussion of the objectives of the study and elaborated in the results and conclusion.

Reviewer 2 Report

The authors designed a coherent experiment to examine their research question surrounding how framing of immigration-related messages, campaigns, and behavioral pleas moderate the moral foundations of participants on their immigration attitudes, voting behavior and behavioral outcomes (i.e., donations).  As a start, this research is well-done and is worth considering for publication, but needs substantial revisions.  In particular, literature is missing from the review; directly related key findings by Nath, Pedriana, Gifford, McAuley & Fulop (2022) ("Examining Moral Foundations Theory through Immigration Attitudes," Athens Journal of Social Sciences, Vol 9 Iss 1) which found political orientation effects to be significantly moderated by alternative moral foundation framing of pro-immigration appeals on immigration attitudes, though whether attitudes are entrenched or persuaded into a new attitude depends on one's place on the political spectrum.  Nath et al's (2022) research associates closely with this study and would best be incorporated into the literature review and the findings. 

As for the method used, it was a creative experimental design.  Regarding explanations around sampling (in particular how they chose to include or exclude participants), I appreciated their clarity.  I wondered however, how results would have differed if they excluded after all those participants who incorrectly answered the math question, Q6, in the MFQ30 measure, since this is how Moral Foundations Theory prefers it, rather than including them. A footnote added on this would be welcome.

Turning to the results section, it begins wonderfully in presentation and with clear explanation of results, but unfortunately, around line 271, drops off and begins to read like researcher notes.  I'd recommend expanding on those short notations and the findings with clear complete paragraphs. 

As another rather important side but easy to fix issue, a p-value of "p<.00" is often noted in this paper which is not the scientific standard.  The standard is to rely on three levels of significance reporting: p<.05; p<.01; or p<.001.  Therefore, for any p<.00, simply write "p<.001" for all of those cases; these should be corrected (for example, it might be this p<.00 is from p<.00003 or similar, thus we default to p<.001). The same goes for Table 2's "p<.02" which instead is best expressed simply as "p<.05" to follow the scientific standard.

Overall, this study appears sound except for some missing literature, reflection on that literature in the review and conclusion, and some problems with the presentations of results and conclusion.  I recommend it be revised and resubmitted for consideration again.

In addition to what is mentioned above which I feel is very important and requires attention before publication, there are also these easier to fix areas:

  • Line 83-83: Please include brief examples with what you mean by: "Compare the expression of care/harm in Ancient Spartan society with that of contemporary California."
  • Lines 92-95 that begins "These loyalty predispositions..." and so on is a very awkward sentence and needs rewriting for clarity.
  • Line 105: "social word" should be "social world".
  • Line 108: This sentence need rewriting: "Here we looked at the effect of these frames on focused on four of these frames."
  • Lines 128-132: The word "study" appears twice in a row (delete one) and again later in the sentence which is awkward, and then the whole sentence is quite long and unclear. Rewrite it.
  • Line 254: "3. Results" I believe needs to go on the next line?
  • Line 271-356: This area in the results section is very incomplete reporting. The language reads more like author notes than a research article ready for publication to be understood by readers. Please expand on these areas.
  • Line 293, for example and elsewhere in the results section, it is not stylistically (or grammatically?) appropriate to begin a sentence with a number unless you spell it out. Since these are large numbers, instead try, "On the moral of care, 695 individuals scored highly."  Make similar changes throughout the results section.
  • Chi-squared is written differently, sometimes as "Chi2", "Chi2 square tests" and "X2".  I recommend simplifying all to be the same: "Chi-squared..." and "Chi-squared tests..."
  • Line 362: Add a space; from "significancebetween" to "significance between"
  • Line 372 onward is another area where Nath et al (2022) applies and may be mentioned.
  • Line 393-394 includes some emotional language which is best left out of scientific reporting.
  • Double check all literature mentioned in the paper is cited in the references, and vice versa.

Reviewer 3 Report

Referee report on "Moral foundational framing and its impact on attitudes and behaviors"

The paper in front of us provides an experimental exercise on the moral foundations theory. The author identifies five moral values and checks how different foundations can affect people’s responses on politically oriented framing. In particular, within the framework of pro-refugee policy, he/ she has showed that individuals who scored highly on care and fairness responded most positively to corresponding moral frames, while framing did not influence attitudes and behavior of individuals who score highly on sanctity and loyalty. This is an interesting exercise that immediately attracts attention, and the author was able to find his/ her own niche within the voluminous literature on framework.

My comments on the paper are below.

  1. Please re-write the Abstract more precisely. Some of the ideas are repeated unnecessarily.
  2. Paragraph 2 in the Introduction, which motivates the story, should be seriously revised. The first two sentences do not suffice to introduce the story of the attitudes toward immigrants. To provide a more balanced description I would suggest after the first two sentences (line 34, p. 2) to add the following:

“Predictably, it has been found that across European countries non-Western and, particularly Muslim immigrants, induce threat perceptions in the host society (e.g. Schneider, 2008; Green et al., 2010; Hjern and Nagayoshi, 2011), and that immigrants of a different race/ethnicity to the native population appear to be perceived as being more likely to benefit from public funds than immigrants of the same race/ethnicity as the native population (Bridges and Mateut, 2014). More generally, it has been shown that the influences of unskilled immigration, differential fertility between immigrants and the local indigenous population, and the incentives for investment in human capital can combine to predict decline of the West (Azarnert, 2010). To deal with the refugee crises, it has also been suggested that, while poorer countries accede to host refugees, richer countries should help provide financial support to those host countries to protect health and well-being of resettled refugees (Azarnert, 2018).”

2a. Start the new paragraph with the third sentence of the current text and discuss it in a bit more detail.

  1. Pages 3 and 4: The specific moral foundations you deal with in the paper should be described in a more scientific way.
  2. Tables should be seriously re-arranged. Not all of them are necessary in the main text. These unnecessary details disrupt the text. In my discipline (economics), we relegate some of them to an Appendix.
  3. Monetary donations

5a. Your description of monetary donations is not clear enough. In some places you write that the donation could be 20% of the experimental earnings, while in other places you mention 0.20 BP. Let me say that 0.20 BP is meaningless and I don’t think that people would care about such amount of money.

5b. Did people in your sample fully understand that it is a Fictional Charity, not a Real one? For example, in “dictator” or other such games experiments, it has been broadly observed that white people, especially females, are often ready to give more if they know that they play against an anonymous, but real refugee or other poor minority person. If you can formulate any way to compare donations to “real” vs. “fictional” charity, it will be interesting. Another alternative is to compare White people’s fictional donations to those of minorities and donations of males to those of female. Otherwise, drop this monetary experiment.

  1. More generally, I would suggest taking the experiment in the direction of the comparison across (a) Gender and (b) Whites vs. minorities. In particular, it would be interesting to see whether minorities are more pro- or against- refugees than White people in general, and, if possible, indigenous British people.
  2. Another interesting further extension could be to check do people really understand what an increase in the number of refugees in the UK by 5% means in terms of people. Given your data, is it still possible to play with the percentage (5%) versus the number of actual people who arrive to Britain?
  3. When on p. 14 (lines 407 – 413) you speak about a possible left-wing bias within your sample, I would suggest to strengthen this point in the following direction: You have more females in your sample and females are broadly deemed to be more left-wing oriented. Do your figures support this popular belief?
  4. Double check the order of the names of the authors as they appear in the text and in the reference list. For example, the order of the authors in lines 66 and 72 in the text is different from that in the references.
  5. Clean the text from the typos and some remaining irregularities.

10a. Don’t use the term “xenophobia” in line 394 (p. 14). It’s too strong in your context.

References

Azarnert, L.V., 2010. Immigration, fertility, and human capital: A model of economic decline of the West. European Journal of Political Economy. 26, 431–440.

Azarnert, L.V., 2018. Refugee resettlement, redistribution and growth. European Journal of Political Economy 54, 89–98.

Bridges, S., Mateut, S., 2014. Should they stay or should they go? Attitudes toward immigration in Europe. Scottish Journal of Political Economy 61, 397–429.

Green, E.G.T., Fasel, N., Sarrasin, O., 2010. The more the merrier? The effect of type of cultural diversity on exclusionary immigration attitudes in Switzerland. International Journal of Conflict and Violence 4, 177–190.

Hjern, M., Nagayoshi, K., 2011. The composition of the minority population as a threat: can real economic and cultural threats explain xenophobia? International Sociology 26, 815–843.

Schneider, S.L. 2008. Anti-immigrant attitudes in Europe: outgroup size and perceived ethnic threat. European Sociological Review 24, 53–67.

Round 2

Reviewer 3 Report

Now I'm saticefied and recommend the paper for publication.

Author Response

We have updated the statistical analyses used for this study. The statistics are now more appropriate and the findings more robust. The main findings have not changed. Full details of the statistics will be re/viewable in the supplementary materials attached to the study.

We have also embarked on a degree of refinement to the writing; again, the content has not changed in meaning (for the most part), and the spirit of the paper remains the same. We have added some additional references too.